# An Empirical Study of SETA Program Sustaining Educational Sector's Information Security vs. Information Systems Misuse

**Binglong Zheng, Daniel Tse \*, Jiajing Ma, Xuanyi Lang and Yinli Lu**

College of Business, City University of Hong Kong, Kowloon, Hong Kong, China;
bingzheng7-c@my.cityu.edu.hk (B.Z.); jiajingma2-c@my.cityu.edu.hk (J.M.); xlang6-c@my.cityu.edu.hk (X.L.);
yinlilu2-c@my.cityu.edu.hk (Y.L.)

\* Correspondence: iswktse@cityu.edu.hk

**Abstract:** Information systems misuse and data breaches are among the most common information security threats at the organisational and individual levels. Security, Education, Training and Awareness (SETA) program can be effective tools in addressing and preventing such risks for sustaining the educational sector's information security, although it is costly to implement and achieves limited results. Several studies have shown that SETA implementation can improve corporate employees' information security protection behaviours. This study adopts the method of quantitative research, deterrence theory with selected perceived cost and information security awareness (ISA) as intermediate variables and explores how SETA programs affect information system abuse on campuses. The results show that implementing the SETA program positively impacts perceived cost and ISA; perceived cost and information security positively impact reducing misuse behaviour of information systems. At last, we provide rationalisation suggestions for individual students and schools to help SETA programs to be better implemented.

**Keywords:** SETA; sustainability; perceived cost; IS misuse; information security awareness

## 1. Introduction

### 1.1. Research Background

With the popularity of the Internet and the acceleration of e-commerce processes, cybersecurity has become a key concern for companies. Information security hazards caused by computer viruses and data leakage can directly affect the competitiveness and reputation of enterprises [1]. In addition, some studies have shown that the losses that come from compromises in information security are huge, and most of the security incidents are caused by human factors within the organisation, such as security breaches brought about by misuse of information systems [2]. Therefore, it has become urgent for organisations to enhance employees' ISA to reduce the impact of such behaviours on business.

With the rapid expansion of smart devices and the continuous development of the Internet, people cannot bypass the Internet and electronic devices for work and life, and at the same time, they cannot avoid information security problems. There has been an insurmountable gap between personal information security and enterprise information security. However, with the popularisation of personal electronic devices and the continuous iteration of intelligent programs, the functions of personal devices have run through people's work, study, and life.

Most studies have focused on implementing SETA in enterprises and its impact on employees' behaviour [3]. However, students are also essential stakeholders that use information systems and are threatened by information security. The epidemic's impact has made it increasingly important for information systems to function well in the campus environment. Misuse and abuse of information systems by students, faculty or technical staff reduces learning or office efficiency and may bring potential threats and losses to the

school. Security education and training for students is one of the effective, sustainable measures to deal with such threats.

In a company, employees need to use the information system and their smart devices to access the company system for their office work [4]. Similarly, many college students must use mobile phone applications to enjoy campus services. They must also use the school's information system, such as the e-learning office, to participate in regular teaching. Numerous studies have shown that "60 to 85 per cent of information security incidents is caused by a lack of knowledge and understanding among people inside the organisation" [5]. Therefore, whether it is for companies or schools, improving users' security awareness of their information systems is necessary. SETA is one of the many means of this problem [6]. Since the birth of SETA as an ideology of information security, its application fields and research areas are concentrated in companies and campuses with information security management problems have long maintained obvious research gaps.

SETA training usually involves various aspects such as general knowledge, awareness and culture of information security, which can further develop the ability of individual members within an organisation to deal with cyber security threats [1]. By implementing SETA programs, students can learn how to deal with cybersecurity breaches and threats and apply them to their daily academic life, such as accessing information via the Internet, sending and receiving emails, and using school information systems to complete tasks. However, the factors through which the implementation of SETA influences individual students' behaviour towards information system misuse need further research.

### 1.2. Research Purpose & Significance

With the impact of the COVID-19 epidemic on education, more students have begun to use electronic devices to participate in teaching and learning interactions. This reflects that the Internet is improving the quality of life and promoting the level of education, whereas some inevitable information security problems exist. Therefore, it is necessary to explore the safety behaviour of students' equipment and improve it.

This study examines how the current SETA program affects students' psychological perceptions and reduces their misuse of information systems to verify whether each factor has a positive impact and significance. Based on the findings, an effective management plan can be developed for universities to improve students' information security behaviours and protect teaching resources from malicious attacks.

This study attempts to carry out quantitative research on the SETA program on the student information system based on various relevant theories, conduct statistical modelling of the collected questionnaire data, and empirically analyse the perceived cost of students under the SETA policy, the impact of information security awareness on its compliance with information security rules or misuse of information systems. This research can provide references and suggestions both for schools and students. On the one hand, it contributes to helping schools knowing how to conduct an effective SETA program to cultivate students' information security awareness and improve information security regulations. One the other hand, it forms the side guidelines for students. It educates them to be active in the school's information security activities, reducing information security threats from the root link of individual behaviour. Furthermore, this study fills the gaps in existing research and lays an excellent theoretical foundation for future studies exploring the SETA program's effects on students. In other words, this study has obvious theoretical significance and practical significance.

### 1.3. Research Framework

Like most quantitative research, this article follows the conventional empirical writing structure: The first part of this article is the Introduction, which summarizes the background, purpose, innovation and design; the second part is a literature review, which introduces various researches and theoretical foundations in the related fields; the third part is research model, which briefly introduces our model building and variable definition;

the fourth part is hypothesis building explains how we develop four hypotheses; the fifth part is data analysis, mainly contains empirical results, including statistical analysis of the questionnaire, reliability and validity analysis and factor analysis; the sixth part is hypothesis testing, which conducted test on our hypothesis and give result; the seventh part is analysis and discussion for results explanation; the eighth part is the limitation and prospect, which discusses our defections and future research; the last part is conclusion and suggestion, which makes summary and gives recommendations.

### 1.3.1. Questionnaire Survey Method

The questionnaire method refers to a process of social research that uses self-administered questionnaires or structured interviews to systematically and directly collect information from a sample of social groups and then analyse the information statistically to understand social phenomena. This study uses quantitative analysis to verify the research hypothesis, design questionnaire items based on the combination of the previous literature and the mature scales of domestic and foreign scholars, and then distribute and return the questionnaire.

### 1.3.2. Empirical Analysis Method

The empirical research method is used to test pre-established research hypotheses or propositions. If the results shown by data analysis are consistent with the expectations of the research hypotheses, the hypotheses are considered valid; otherwise, the original hypotheses should be rejected. In this study, according to the general deterrence theory, we construct a theoretical model and propose four hypotheses that affect students' information systems misuse. We collect sample data through a questionnaire and use SPSS to process the obtained data and draw conclusions.

## 2. Literature Review

Our study focuses on the mechanism of the SETA program towards information system misuse by analysing the intermediate function of perceived costs and information system awareness. Therefore, we do a comprehensive literature review of each concept and its theoretical underpinning.

### 2.1. Information Security

Information exchange has become more convenient with the continuous development of technology, hardware facilities and the Internet. "The information system is a set of software, hardware, data, people, procedures, and networks that enable the organisation to use information resources. Moreover, information security should be implemented into every central system in an organisation" [7]. In recent years, more and more people and organisations have seen the importance of ISA.

Information security must be protected by technical means, but technical methods are easily invalidated. Losses can be effectively reduced through deeper policy solutions to manage information security [8].

In one report, 38% of data breaches are caused by losing paper files, 27% are related to incorrect storage on mobile devices and only 11% are due to hacker attacks. Therefore, information security management should be managed more comprehensively, most notably by controlling human-generated errors [9].

### 2.2. Security, Education, Training, and Awareness

Employee errors may be the top threat to information assets, so it is worth taking a project into implementation to combat this threat. "Security education, training, and awareness (SETA) is a managerial program designed to improve the security of information assets by providing targeted knowledge, skills, and guidance for an organisation" [7]. There are various ways to conduct information security management, including technical

management, policy, and employee management programs. The SETA program is a relatively low-cost protection mechanism with potentially high investment returns [10].

The SETA program is closely related to the policy of information security. The policy of Information security contains two parts, one is the effectiveness of the policy, and the other is related to the implementation of the policy. Therefore, even if a viable policy is developed, information security still can only be guaranteed if the performance is satisfactory. Thus, an effective SETA program allows the organisation to protect its information.

Training in information security plays a crucial role in the enterprise. On the one hand, such measures are more effective in raising awareness of information security within the organisation. On the other hand, such training can push internal staff to operate, violating the rules [9]. "SETA implementation conveys impact to the reorganisation in terms of raising ISA, understanding the importance of organisational information security, and training individuals to assume information security roles" [11]. SETA can provide employees with security knowledge and skills through continuous efforts to enable them to deeply understand why they need security protection and raise their awareness of security issues leading to ISA [12]. "Organizations provide security, education, training and awareness programs through the use of effective training techniques to educate employees on how to make proper security information decisions", thereby reducing poor security behaviour of employees towards organisational information resources [13].

### 2.3. Information Security Awareness

A quantitative model has been developed to measure information security incidents in organisations caused by human error related. The experimental results show that more than half of the information security incidents have their root cause in human error [14]. This reflects that most information security incidents in an organisation are theoretically avoidable. Avoiding such cases will significantly improve the organisation's efficiency and reduce additional expenses. By fostering and building awareness of information security among people, the effectiveness of information security policy implementation will be significantly enhanced [9].

For individuals, increased ISA can effectively prevent mistakes that could lead to information leakage. According to the study, the more threats respondents perceive, the more protective their behaviour becomes. Those who are trained are more aware of the issue than those who are not trained, and it can be seen that the former behaves more protectively. Among all respondents, students aged 18–30 are the most exposed group to information security risks [15].

Therefore, in this era of rapid Internet development and new technologies with unknown risks, students need to develop and build ISA. This will be reflected when they step into the workplace.

### 2.4. General Deterrence Theory

General deterrence theory (GDT) is a famous concept in the legal system and criminology that proposes using punishment as a threat to deter people from committing crimes. General deterrence dissuades people from imitating the perpetrator by publicly condemning or punishing them [16]. Based on general deterrence theories, Lee et al. develop an integrative model of computer abuse and point out that general deterrence factors are security awareness, security policy and physical security system. They found that organisations with solid deterrence factors show a higher sense of self-defence than those with weak deterrence factors regarding computer abuse [17]. Likewise, D'Arcy et al. extend the GDT model and conclude that there is a deterrent effect of security policy, SETA program and computer monitoring "on IS misuse intention. This effect is achieved indirectly" through the intermediate factor of perceived sanctions [2].

Based on the above review, GDT can be used in our research. The basic principle of GTD is SETA program serves as a deterrent mechanism to reduce the incidence of

information system misuse by generating perceived cost awareness and raising information security awareness.

### 2.5. Perceived Cost

Perceived cost mostly appears in consumer purchases, which presents the sum of expenses that customers have in the actual consumption process, involving time, money, physical strength, psychology and other costs. Gradually, perceived cost permeates various research fields and has been confirmed to be the influencing factor which leads to changes in individual behaviour. Salim et al. prove that perceived cost plays a vital role in facilitating the formation of an organisation's willingness to adopt blockchain technology [18]. Benet et al. verify that the user's financial cost is the main obstacle to IOT adoption. Because of the high perceived cost, end users will not choose IOT even if they trust this technology, which would bring high perceived benefits [19]. Saedi's research finds that perceived cost harms mobile payment [20].

Spence states that every activity would generate costs and benefits while the individual is willing to avoid a task whose cost is much higher than the benefit. Firstly, Spence comes up with three types of costs: Effort cost—How much effort it takes and whether it is worth it. Opportunity cost—The energy, time and money of conducting one activity take away from another. Emotional cost—The emotional and social costs of expected anxiety and failure generated by the pursuit of completing tasks [21]. Based on the fundamental classification, researchers have proposed detailed types of costs in line with different research topics. To clarify the relationship between customers and green brands, Papista and Krystallis subdivide the perceived cost into effort, evaluation and time [22]. Prashant classifies the perceived costs as time and financial costs to explore their impact on students' intentions to adopt online classes.

Hence, our research could apply the perceived cost to the GDT model. After suffering information system misuse, students may suffer a list of unfavourable costs, such as the cost of having accounts banned if they help students from other schools download materials, the cost of failing a course if they use a system leak to change their grades and the cost of having their computers attacked if they mistakenly click on phishing emails. All in all, our goal is to investigate the intermediate role of perceived cost in the impact of the SETA program on information system misuse.

### 2.6. Summary

Although many previous informative types of research lay a solid foundation for our analysis, their target objects range from supervisors to employees. They cannot quantify the SETA program's function from students' perspective. Besides, the studies which emphasised the deterrent effect of the SETA program are limited even if the SETA program's advantages have been promoted for a long time. Thus, our research focuses on the impact of the SETA program on information system misuse from the perspective of students and through the indirect effect of perceived cost and ISA.

## 3. Research Model

### 3.1. Model Building

This study investigates whether the SETA program implemented on campus in the information security environment impacts students' psychological cognition and information system misuse behaviours and the mechanism of the whole impact process. The general deterrence theory (GDT) is an essential theoretical framework. Evaluating the deterrent capacity of the SETA program is especially important. Although the advantages of the SETA program have been widely publicised, there is limited empirical research on its deterrent effect. In our study, the basic principle of GDT is, through propagating the severe consequences of information system misuse (e.g., financial loss and time loss to individuals and schools), the SETA program serves as a deterrent mechanism to reduce the incidence

of information system misuse by increasing students' perceived cost and awareness of information security.

As shown in Figure 1, our research model integrates the SETA program, the perceived cost, information security awareness and the reduction of information system misuse. This model is an extension of GDT, which considers the SETA program as the antecedents, and perceived cost and information security awareness are the intermediate factors. That is, the SETA program indirectly influences the reduction of information system misuse through its effect on students' psychological cognition.

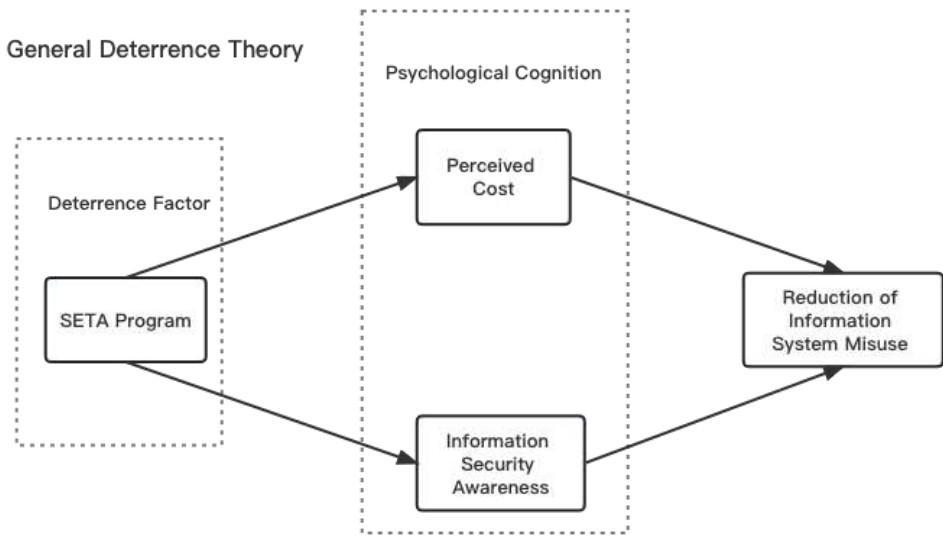

**Figure 1.** Research Model.

### 3.2. Variable Definition

As shown in Table 1, based on the research content and theoretical model, we list the definitions of relevant variables at the level of deterrence factor, psychological cognition and individual behaviour in the influence mechanism of information systems misuse.

**Table 1.** Research variables definition.

| Aspect | Variable Name | Variable Definition |
|---|---|---|
| Deterrence Factor | SETA Program | The school conducts information security awareness, training and education for students, focusing on the severe consequences of information systems misuse and basic knowledge. |
| Psychological Cognition | Perceived Cost | After receiving information security training and education, students will generate a cognition that misuse of information systems may bring many unfavourable costs, such as the cost of having accounts banned if they help students from other schools download materials, the cost of failing a course if they use system leak to change their grades, and the cost of having their computers attacked if they mistakenly click on phishing emails. |
| | Information Security Awareness | After receiving information security training and education, students will automatically be aware of protecting personal information, system security and defending against attacks. |
| Individual Behavior | Reduction of Information System Misuse | After receiving information security training and education, students will unconsciously reduce the behaviour of information system misuse. For instance, students may improve the ability to identify phishing software, not lend personal accounts to others or help others download materials, do not use school computer room equipment to log in to unsecured websites. |

## 4. Hypotheses Building

### 4.1. The Impact of the SETA Program on Psychological Cognition

Previously, the GDT model has been extended with security countermeasures and other factors by the research on the SETA program area [2]. According to the general deterrence theory, the SETA program can be considered a deterrent factor that promotes users to generate the perceived cost and information security awareness. Perceived cost and ISA can be analogous to short-term memory and long-term memory in psychology. They both belong to the scope of psychological cognition. The difference is that perceived cost is temporary and short-term, and ISA is persistent and long-term. In our model, we will test the relationship between the SETA program and perceived cost and the relationship between the SETA program and ISA.

#### 4.1.1. SETA Program and Perceived Cost

As mentioned above, perceived cost arises from consumer psychology. It is mainly reflected in the cost that people perceive on the spot when they take action, which is a relatively advanced link in the psychological cognition chain. Explained in business terms, this is external information. Some other studies that incorporate perceived cost see perceived cost as part of social exchange theory in psychology [23]. In addition, the SETA program is a general factor that affects users' psychological cognition of information security. Suppose the organisation uses the SETA program to educate and train their users to help them acquire information security knowledge and operational principle and improve their information security awareness. In that case, the users will know the consequences that some detrimental behaviours will bring to the organisation and themselves.

In this study, the SETA program mainly includes information security courses offered on campus, publicity about information security, assessment of information security, penalties for information security, etc. Assuming students are about to engage in misuse behaviours, they face the cost of doing them. If students are accepted into the SETA program, they will consider punishment and the consequences in the cost of engaging in misuse behaviour when they encounter these situations.

Accordingly, hypothesis H1 is proposed:

**H1.** *The SETA program has a positive impact on students' perceived cost.*

#### 4.1.2. SETA Program and Information Security Awareness

ISA is a kind of knowledge, which is the end of the psychological cognition chain formed by people after processing information. One study concluded that "lack of proper training and oversight is a contributing factor behind many information security breaches" [24]. Security, education, training and awareness (SETA) are crucial to enhance and improve employees' information security behaviour. It is often perceived as futile to educate users, firstly because security issues are complex and varied, and it is impossible to ensure the effectiveness of what is taught, and secondly, because it is felt that information security is often perceived as secondary and is not given sufficient attention. However, in [3]'s model, SETA program is delivered to users and internalized into their implicit information security perceptions.

Corresponding to the school scenario, if students participate in SETA programs organised by the school, including but not limited to listening to information security lectures, participating in information security surveys, learning about information system security usage codes, seeing information security posters posted on campus, etc., all these information will be combined into students' awareness of information security. Moreover, often, the more security knowledge or rules they are willing to learn, the more vital and more positive the ISA they internalise in their minds. More forms and perspectives of SETA training will be more helpful to make up for students' lack of information security knowledge and produce positive and effective preventive effects for them when they face the security risks brought by the information of computer systems in the future.

Accordingly, the hypothesis H2 are proposed:

**H2.** *The SETA program has a positive impact on students' ISA.*

*4.2. The Impact of Psychological Cognition on Information System Misuse*

4.2.1. Perceived Cost and Reduction of Information System Misuse

As mentioned, we include perceived cost as a primary psychological cognition variable from consumer purchases research. The perceived cost originates from studying consumer psychology of customers, but it can also be extended to research in various fields. In mobile commerce, the perceived cost can be "the cost involved in the mobile commerce service, including transaction cost, device cost, application download cost, and access cost" [25]. In this study, perceived cost is a variable between the SETA program and the reduction of information system misuse, focusing on students' perception and understanding of punishment after information system misuse. Similarly, in the field of information security, perceived risks and perceived sanctions, which are similar to perceived costs, have already been applied in some research [2]. Perceived cost can also be linked to the SETA program above and extended in the context of this study [26]. Thus, the perceived cost of information security can include the cost of consequences after information system misuse and the cost by penalty.

On campus, assuming that students are on the edge of engaging in information system misuse behaviours, they realize that the costs of engaging in these behaviours include associated harmful consequences and penalties, and they are likely to choose not to engage, thereby reducing the occurrence of related behaviours.

Accordingly, the hypothesis H3 is proposed:

**H3.** *The perceived cost positively impacts students' reduction of information system misuse.*

4.2.2. ISA and Reduction of Information System Misuse

The consciousness often manipulates and eventually maps onto people's actual behaviour. With a complete understanding of security needs and support, people can embody the information they receive in their behaviour and security education will be effective [27]'s study demonstrated that increasing personnel awareness of security issues is considered the most cost-effective implementation within an organisation because increased security awareness allows employees to focus on improving technical operations, which leads to eventual security behaviour in a more desirable direction. [3]'s study of existing research on information security suggests that practitioners need a security awareness theory to explain the expected outcomes of specific awareness initiatives and why this happens. In their model, users receive security awareness training and security information is passed on and internalised as implicit knowledge for the user.

For the research direction of this study, ISA is an intermediate psychological variable, we need to focus on the outcome of ISA and how it will guide behaviour. For instance, if students receive a phishing email and have good ISA to identify and prevent phishing emails, this will guide them to implement the behaviour of deleting or reporting phishing emails. The better a person's awareness of information security events, the better he or she will be able to anticipate risks, enabling him or her to quickly identify threats and react to maximised information assets from being violated.

Accordingly, the hypothesis H4 is supposed.

**H4.** *ISA positively impacts students' reduction of information system misuse.*

## 5. Data Analysis

*5.1. Questionnaire Design*

In designing the questionnaire's content, in addition to basic information such as gender and education, we focus on four dimensions of SETA, perceived cost, ISA and information system abuse to complete the data collection.

In the questionnaire panel about SETA, we borrow the research dimensions and the surveyed questionnaire topics for the SETA program from the articles of Stephanou and

D'Arcy et al. (2009) [8]. Most questionnaires address the education of employees within the organisation on computer security responsibilities and the consequences of informing employees of unauthorised access to or modification of computers [3]. We cover the above elements and add computer security responsibility promotion, which is more relevant to students' identity and more in line with the school environment, as a research question to study better how the SETA program affects the student population.

In designing the questions for the perceived cost section (Section 2.5), we draw on Prashant's article on the perceived cost of financial and time barriers to student participation in online classes during COVID-19 and add the perceived cost of malicious exploitation of system vulnerabilities or access to information when using information systems on campus to design the questions from the student's perspective [28].

When investigating information security awareness, we refer to the general ISA mentioned by [29], i.e., knowledge of potential security threats and their negative consequences and an understanding of people's concerns about information security and the risks they pose [29]. In order to be more relevant to students' life experiences and characteristics, we explain the ISA in the questionnaire topics more specifically, such as the habit of changing passwords regularly and not using the Internet in public places, as well as the behaviour of trying to avoid downloading software from third-party websites.

Since students may engage in information system abuse behaviours or have an existing history of abuse when designing the questions for abuse, we distribute mock phishing emails to students to examine their abuse behaviours. Simple quizzes are also included to collect students' abusive behaviours when using campus devices.

### 5.2. Descriptive Statistical Analysis

We adopt the questionnaire survey method and utilise the WJX (WenJuanXing) questionnaire platform to design and store the questionnaire. The questionnaires are mainly sent to student groups on social media through QR codes and questionnaire links. In order to achieve the broad, universal and random distribution characteristics of simple random sampling, we recruit students from all education levels and different schools to minimize the impact of limitations in education level and region on the results of the questionnaire. Considering the different levels of exposure to and use of information systems, the more educated student groups tend to be more familiar with the SETA program. The results of the questionnaire collection also show that the student population willing to be inter-viewed and complete the questionnaire is mainly concentrated at the undergraduate and graduate levels. After issuing the questionnaire, 150 data have been collected. We eliminate the unqualified data with the same answers and short response time, the remaining 131 valid data can be further used, and the effective utilization rate reaches 87%. By organizing and counting the valid data, descriptive statistical analysis is organized as shown in Table 2.

**Table 2.** Descriptive statistical analysis of the sample.

| Characteristic | Category | Number | Proportion |
|---|---|---|---|
| Gender | Male | 73 | 55.7% |
| | Female | 58 | 44.3% |
| Education level | Junior college | 4 | 3.0% |
| | Undergraduate | 64 | 48.9% |
| | Postgraduate (Master) | 60 | 45.8% |
| | Doctor (PhD) | 3 | 2.3% |

Table 2 shows that the difference in gender distribution in the survey sample is within an acceptable range, with a difference of 11.4% between males and females. Gender records basic demographic information. It proves that our survey is conducted randomly and has no gender preference. The effect of gender on the survey results is excluded. In education level, undergraduate, postgraduate and doctoral students take up a large percentage, the total proportion is 97%. The junior college students, which take a percentage of 3.0%, can be

seen as the same as university students in some degrees. Because our goal is to investigate the impact of the SETA program on students' information system misuse, which determines the survey should be done in a campus environment. Hence, the percentage of education level is consistent with our target group and effectively supports future findings.

### 5.3. Reliability Analysis

The reliability analysis can ensure that the results are valid and reliable. In this study, we propose to use Cronbach's alpha coefficient, which can deal with the internal consistency coefficient and ranges from 0 to 1. The higher the coefficient, the higher the reliability of the scale. We use SPSS to process the data, and Cronbach's alpha coefficient values for each variable scale are shown in Table 3.

**Table 3.** Reliability analysis result.

| Scale | Variable | Cronbach's $\alpha$ Coefficient | |
| --- | --- | --- | --- |
| Deterrence factor | SETA program | 0.725 | 0.725 |
| Psychological cognition | Perceived cost | 0.855 | 0.889 |
| | Information security awareness | 0.792 | |
| Individual behaviour | Reduction of information system misuse | 0.756 | 0.756 |
| Overall | / | / | 0.814 |

The Cronbach's alpha coefficients for all scales are more significant than 0.7. The deterrence factor scale and individual behaviour scale ranges from 0.7 to 0.8, 0.725 and 0.756, and the psychological cognition scale is more significant than 0.8, which is 0.889. Most importantly, the Cronbach's alpha coefficient for the overall scale reaches 0.814. Therefore, the scale's reliability is good, and the variables are measured with solid reliability.

### 5.4. Validity Analysis

This study uses the KMO test and Bartlett's spherical test to verify the construct validity of the scale and the correlation between the variables. The KMO value is generally between 0 and 1, and the closer the KMO value is to 1, the stronger the correlation between the variables. In social science research, a KMO value greater than 0.5 and a $p$-value less than 0.05 indicate good validity. This study uses SPSS to conduct the KMO test, and Bartlett's spherical test on the sample data, and the results are shown in Table 4.

**Table 4.** KMO test and Bartlett's spherical test.

| Scale | KMO | Approximate Chi-Square | Degree of Freedom | Significance |
| --- | --- | --- | --- | --- |
| Deterrence factor | 0.671 | 89.086 | 3 | 0.000 |
| Psychological cognition | 0.834 | 380.123 | 15 | 0.000 |
| Individual behaviour | 0.679 | 98.291 | 3 | 0.000 |
| Overall | 0.840 | 810.596 | 66 | 0.000 |

From the test results, the overall KMO value of the scale is 0.840, which is considered strong validity. The KMO values for the deterrence factor scale, psychological cognition scale and personal behaviour scale are 0.671, 0.834 and 0.679, which all meet the requirement. In addition, the significance of the above three scales and the probability of significance of the overall scale are less than their significance levels ($p < 0.05$). Hence, the sample in this study is measured with good validity.

## 6. Hypothesis Testing

### 6.1. Linear Regression

"Linear regression assumes a linear relationship between the dependent and independent variables", and the dependent variable can be obtained by linearly superimposing the

independent variables [30]. The most common method is the ordinary least squares (OLS). We use the OLS test to verify the influence relationship and significance level between the independent and intermediate variables and the intermediate and dependent variables.

### 6.1.1. Rationalization of the Existing Intermediate Variables

The model assumes that there are intermediate variables in the effect of implementing the SETA program on the student population. In addition to the test of four necessary paths, we also test the path from SETA Program to the reduction of information system misuse, and the results are shown in Table 5. It is found that the OLS model has failed to pass the t-value, $p$-value and F-value tests. Indicating at a significance level of $p = 0.05$ that implementing the SETA program cannot directly impact the reduction of information system misuse by students. In this case, it is quite correct that we set intermediate variables in transmitting the impact process from implementing the SETA program to reducing information systems.

**Table 5.** OLS result for SETA impact on misuse in university.

| | Unstandardised Coefficients | | Standardised Coefficients | t-Value | $p$-Value | $R^2$ | Adjusted $R^2$ | F |
|---|---|---|---|---|---|---|---|---|
| | B | Std. Error | Beta | | | | | |
| alpha | 3.126 | 0.409 | - | 7.636 | 0.000 ** | 0.026 | 0.016 | 2.58 |
| SETA | 0.171 | 0.106 | 0.16 | 1.606 | 0.111 | | | |

** $p < 0.01$.

### 6.1.2. The Impact of the SETA Program on Perceived Costs

This paper argues that there are two mediating variables, perceived cost and information security awareness, within transmitting influence. Implementing the SETA Program influences these two variables, which in turn affects the misuse of information systems. The OLS test is first conducted to examine the effect of implementing the SETA Program on perceived cost.

Like result shows in Tabel 6, it is clear that implementing the SETA program significantly positively influences students' perceived cost (β = 0.358, $p < 0.001$), which means that hypothesis H1 is held.

The results in Table 6 illustrate that SETA Program has a positive effect on perceived cost. More importantly, it proves that implementing the SETA Program can modify the perceived cost of the student body towards information systems. This helps the student body to construct a correct perception of information systems and establish a correct operational concept.

**Table 6.** OLS result for SETA impact on perceived cost in university.

| | Unstandardised Coefficients | | Standardised Coefficients | t-Value | $p$-Value | $R^2$ | Adjusted $R^2$ | F |
|---|---|---|---|---|---|---|---|---|
| | B | Std. Error | Beta | | | | | |
| alpha | 2.459 | 0.357 | - | 6.89 | 0.000 *** | 0.128 | 0.121 | 18.838 |
| SETA | 0.403 | 0.093 | 0.358 | 4.34 | 0.000 *** | | | |

*** $p < 0.001$.

### 6.1.3. The Impact of the SETA Program on ISA

Table 7 shows the results of the OLS model with ISA as the dependent variable and SETA Program as the independent variable is used to verify whether there is an effect of the implementation of the SETA Program on ISA. The result in Table 7 shows that implementing the SETA program significantly impacts the increasing ISA of students (β = 0.405, $p < 0.001$), which means hypothesis H2 is held.

**Table 7.** OLS result for the impact of the SETA program on ISA.

| | Unstandardised Coefficients | | Standardised Coefficients | t-Value | *p*-Value | R$^2$ | Adjusted R$^2$ | F |
|---|---|---|---|---|---|---|---|---|
| | B | Std. Error | Beta | | | | | |
| alpha | 2.499 | 0.274 | - | 9.002 | 0.000 *** | 0.164 | 0.158 | 25.164 |
| SETA | 0.358 | 0.071 | 0.405 | 3.794 | 0.000 *** | | | |

*** $p < 0.001$.

Implementing the SETA Program is a way to increase ISA levels in the student population. The adjusted R-squared (0.158) shows that the ISA levels of the student body are due to various factors. This study speculates that this may be related to the student group's education level and past experiences and knowledge of information technology received. It will not be verified too much here.

### 6.1.4. The impact of Perceived Cost, ISA on Reduction of Information System Misuse

Above testing results reflect the impact of SETA programme to perceived cost and ISA separately. Furthermore, it have to test how effective perceived cost and ISA contribute to reduction of information systems. Table 8 shows that the perceived cost (β = 0.441, $p < 0.001$), as well as the ISA (β = 0.279, $p < 0.001$), have a significant impact on the reduction of information system misuse. Furthermore, the adjusted R-squared (0.402) provides strong evidence that the perceived cost and ISA can largely explain the phenomenon of reductions in information system misuse. Therefore, hypotheses H3 and H4 are validated simultaneously.

**Table 8.** OLS result for the impact of perceived cost and ISA on reduction of information system misuse.

| | Unstandardised Coefficients | | Standardised Coefficients | t-Value | *p*-Value | R$^2$ | Adjusted R$^2$ | F |
|---|---|---|---|---|---|---|---|---|
| | B | Std. Error | Beta | | | | | |
| alpha | 0.694 | 0.346 | - | 2.003 | 0.047 * | 0.411 | 0.402 | 44.384 |
| Perceived Cost | 0.431 | 0.081 | 0.441 | 5.339 | 0.000 *** | | | |
| ISA | 0.348 | 0.103 | 0.279 | 3.383 | 0.001 ** | | | |

* $p < 0.05$ ** $p < 0.01$ *** $p < 0.001$.

### 6.2. Path Analysis

"Path analysis is an extension of the multiple linear regression model" [31]. Path analysis can be used for hypothesis testing when there are interactions between the independent and dependent variables. Meanwhile, path analysis requires testing of the independent variables as well as the mediating variables, which is a step-by-step process.

The model proposed in this essay has several modules and influencing factors. Therefore, the path analysis is chosen to show the influence degree between the variables clearly. This essay uses SPSS as a modelling tool to test the hypotheses of the proposed model.

As shown in Table 9, the path coefficients and some indicators of the constructed structural equation model are obtained using SPSS. From the data in the table, it can be seen that SETA Program has a significant positive effect on perceived cost with a standardised path coefficient of 0.358, SETA Program has a significant positive effect on ISA with a standardised path coefficient of 0.405, the perceived cost has a significant positive effect on the reduction of information system misuse with a path coefficient of 0.465, and ISA has a significant positive effect on the reduction of information system misuse with a path coefficient of 0.295. In other words, as shown in Table 10, all four hypotheses hold true.

**Table 9.** Model path coefficient result.

| Path | Std. Estimate | S.E. | C.R. | *p* |
|---|---|---|---|---|
| SETA Program << Perceived Cost | 0.358 | 0.092 | 4.376 | *** |
| SETA Program << ISA | 0.405 | 0.071 | 5.055 | *** |
| Perceived Cost << The Reduction of Information System Misuse | 0.465 | 0.067 | 6.479 | *** |
| ISA << The Reduction of Information System Misuse | 0.295 | 0.085 | 4.106 | *** |

*** represents it is significant at the 0.001 level.

**Table 10.** Hypothesis testing result.

| Hypothesis | Content | Result |
|---|---|---|
| H1 | The SETA program has a positive impact on students' perceived cost. | Hold |
| H2 | The SETA program has a positive impact on students' information security awareness. | Hold |
| H3 | The perceived cost positively impacts students' reduction of information system misuse. | Hold |
| H4 | ISA positively impacts students' reduction of information system misuse. | Hold |

A comprehensive model with the standardised path coefficient and significance level is shown in Figure 2, combining the linear regression result and the hypothesis test result.

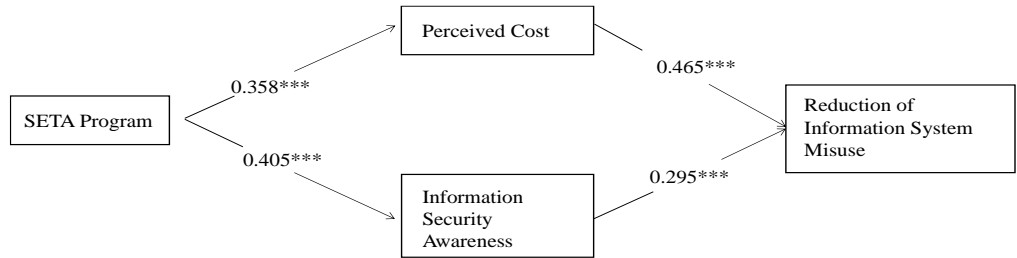

**Figure 2.** Path Analysis Result. *** *p* < 0.001.

## 7. Results Analysis and Discussion

### 7.1. Hypothesis Analysis

#### 7.1.1. Effect of SETA on Perceived Cost

As expected, the H1 has been supported (β = 0.358, *p* < 0.01), which indicates that a campus SETA program is more likely to improve students' perceived cost. Specifically, suppose the school implements the SETA program, including offering information security public and general courses, and promoting relevant knowledge and policies. In that case, students will better understand information security and the consequences and penalties of information system misuse. The SETA program increases the perceived cost of students before engaging in information system misuse cost behaviours by conducting education and training on information security and promoting an information security culture within the organisation.

#### 7.1.2. Effect of SETA on ISA

The quantitative analysis of the questionnaire data also proves that the implementation of the SETA program positively impacted the increase of ISA among students (β = 0.405, *p* < 0.001), i.e., hypothesis H2 holds. This implies that implementing the SETA program can compensate for and increase the level of information security in the student population in cognitive terms. Students with experience with the SETA program, i.e., have been exposed

to information security activities conducted by the school through different forms and tend to show higher risk prevention awareness and better security usage habits. In addition, the adjusted R-squared (0.158) indicates that various factors influence the ISA level of the student population, and the implementation of SETA programs is only one of the influencing factors. This is because students' security awareness can come from many sources and is strongly related to personal factors. For example, some people knowledgeable about computer security and have a risk-averse tendency will perform well on the ISA measure. However, this result may not be due to their security awareness training.

### 7.1.3. Effect of Perceived Cost on Misuse

H3 has been supported ($\beta = 0.441$, $p < 0.01$), which means that when students are about to engage in information system abuses, they will reduce the occurrence of such behaviours if they realise that the harm and departure from them will increase the cost of their behaviour.

### 7.1.4. Effect of ISA on Misuse

The data in the hypothesis testing section (Section 7) ($\beta = 0.279$, $p < 0.01$) supports the hypothesis of H4 that students' ISA has a positive impact on reducing information system misuse. To be more specific, when students face an unknown information security threat and need to make a judgment in using information systems, reasonable ISA can help them avoid risks and reduce misuse or abuse of information systems. ISA can help them identify risks more quickly, guide them to make the correct behaviour and reduce the damage caused by misuse of information systems.

### *7.2. Synergistic Effect*

Combined with the hypothesis testing in Section 6, SETA significantly affects perceived cost and ISA but not misuse, which supports our selection of perceived cost and ISA as intermediate variables between SETA and misuse. We refer to perceived cost and ISA as psychological and cognitive variables, so the two hypotheses H1 and H2 can also be referred to as the impact of SETA on psychological and cognitive variables. Furthermore, according to the verification of H3 and H4, perceived cost and ISA have a significant impact on misuse, so it can be said that psychological and cognitive variables impact misuse. The psychological and cognitive variables are two intermediate variables between SETA and abuse, so it can be inferred that perceived cost and ISA have a synergistic effect on abuse, and the psychological and cognitive variables that cause this synergistic effect are affected by SETA. Therefore, it can be considered that SETA has an indirect effect on abuse between two psychological and cognitive variables, perceived cost and ISA.

## 8. Advice to Universities Applications

### *8.1. Expanding the Scope of the SETA Program*

For university management, the misuse of information systems by students can cause a loss of teaching resources, which are in great demand for protection. This study reflects that the SETA program effectively avoids information system misuse. Therefore, universities should expand the SETA program's publicity channels from various aspects.

The first is text campaigns. The university can print information security leaflets and information system user manuals and distribute them to students to enhance their ability to use information systems safely.

The second is multimedia content guidance. Universities can promote instructional videos on the safe use of information systems through social media platforms and the Internet.

The third is information security education and training. The university can organise information security education and training for new students to help them promptly build information security awareness. To improve the effectiveness of the SETA program, we suggest that the design of the SETA program can be transformed into a SETA artefact. It is found that visual-based gamified SETA artefacts are the most effective type of

SETA program [32]. Therefore, universities can use visual-based gamified SETA artefacts, such as cartoons, comics and short movies, to help students better understand information security knowledge. In this case, students' information security awareness can be improved effectively.

### 8.2. Checking the Effectiveness of SETA Project Implementation

This paper proves that the SETA program does not directly reduce the incidence of information system misuse, and it is essential to check its effectiveness.

On the one hand, the university's Information Systems Management Office can assess the effectiveness of the SETA program at a technical level by conducting routine checks on educational resources.

On the other hand, the effectiveness of the SETA program could also be evaluated by having interview surveys with students to investigate whether their ISA has increased.

### 9. Conclusions

During the pandemic, people have to shift their conversations from offline to online, profoundly affecting later communication. During this period, universities deposit many of their teaching resources on online databases to minimise the impact on the quality of teaching and learning. For university students, resources may be intentionally destroyed or unintentionally leaked during this transformation process, which is a loss of educational resources. Therefore, it is essential to research how to minimise the loss.

In controlling the loss of information resources, implementing the SETA program to the relevant personnel is a widely spread practice. To determine how the SETA program mitigates and prevents the risk of resource loss, we designed a questionnaire to measure university students' perceived cost, information security awareness, and the level of information security education environment and information system misuse.

Based on the questionnaire results, we designed a model for path analysis and constructed an OLS model. The result shows that the SETA program does not directly affect college students' information system misuse behaviour. However, the SETA program indirectly controls it by affecting its perceived cost and information security awareness. In other words, judging a SETA program by whether or not it is implemented without considering its effectiveness may lead to conclusions contrary to reality.

The group reflected in the results is more than just college students. As fresh graduates have just entered the workplace, they do not have a high level of understanding of compliance, which may lead to the leakage of information resources. Therefore, it is essential to provide regular information security training to college students and fresh graduates to raise their awareness of information security to control the loss of information resources.

**Author Contributions:** All authors have shared an equal workload in all areas. All authors have read and agreed to the published version of the manuscript.

**Funding:** This research received no external funding.

**Institutional Review Board Statement:** Ethical review and approval are waived for this study since we conducted the questionnaire survey in Hong Kong and strictly abided by the Personal Data (Privacy) Ordinance (PCPD) in the Data Privacy Law of HKSAR by declaiming the start of our questionnaire survey, which is 'This survey is anonymous, only for our research study this time, you are not compulsory to participate in this survey. This questionnaire survey research depends entirely on your voluntary help, and all your information will be kept strictly confidential and cleared after the survey. We would be very grateful if you could take a few minutes to participate in this survey!

**Informed Consent Statement:** Informed consent was obtained from all subjects involved in the study.

**Data Availability Statement:** Not applicable.

**Conflicts of Interest:** The authors declare no conflict of interest.

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
