# Peer review of "An Empirical Study of SETA Program Sustaining Educational Sector’s Information Security vs. Information Systems Misuse"

_sustainability, doi:10.3390/su151712669_

Round 1
Reviewer 1 Report
The paper presents a detailed study of the impact of the SETA program on educational institutions. The authors have presented the research methodology and hypotheses clearly and followed it with a strong empirical analysis. The study focuses on how GDT is applicable in this case and how the concept of perceived cost impact the student's behavior in the setup. On the other hand, the hypotheses in the paper seem to be very straightforward and do not provide any unexpected or surprising results. One of the major tools used in the paper to do the empirical study is a survey using the questionnaire. It is not clear how many students were asked to fill out the survey (whether it was the 150 students or more people were asked but didn't fill out the survey). The current sample size may be considered too small, but the results are still very predictable. However, the linear regression model is interesting and the weighted graph in the path analysis reesult is an interesting takeaway. The manuscript is clearly written but can be more precise. There are a lot of areas where the concepts are repeated, so the authors can improve the quality of the paper by avoiding such instances and making it more crisp. Furthermore, the authors should stick to one referencing format as advised by the journal's guidelines and follow it throughout the paper.
Author Response
Thanks for the reviewer’s comments. We have addressed the problems as follow:
One of the major tools used in the paper to do the empirical study is a survey using the questionnaire. It is not clear how many students were asked to fill out the survey (whether it was the 150 students or more people were asked but didn't fill out the survey). The current sample size may be considered too small, but the results are still very predictable.
Since our questionnaire is publicly available on the web, it is not clear that the specific information about how many people access and browse the questionnaire. The collected 150 sample data can prove that the number of people who see the questionnaire and fill it out. In addition, we provide a more detailed description of the group of people who received the questionnaire, which covers not only all educational stages but also schools in different regions. This minimizes the error and further supports the fact that we collected a representative sample.
There are a lot of areas where the concepts are repeated, so the authors can improve the quality of the paper by avoiding such instances and making it more crisp.
Upon inspection, we find that there are duplicate conceptual narratives for SETA, perceived cost and ISA in chapter 2 Literature Review and chapter 4 Hypotheses Building. Hence, we remove the duplicate content and move the valuable introduction of these concepts from Chapter 4 to Chapter 2. Besides, the original content of 4.1.2 and 4.2.2 are too long, we revise to make them consistent with 4.1.1 and 4.2.1.
We have also moved part of 1.3 to 1.1 and also combine Research Purpose and Significance into one section.
Stick to one referencing format
Done. We have modified the referencing and now it complies with the journal’s guidelines.
Reviewer 2 Report
The authors seek to present an Empirical Study of the SETA Program Sustaining the Educational Sector’s Information Security vs Information Systems Misuse. Following are some of my suggestions.
· The references need to be re-arranged in the proper sequence. Currently, the references are not in a proper format. I believe it is against the journal guidelines.
· The paper organization (road map) should be clarified for better structure and flow. Currently, the paper organization is missing.
· The motivation behind the research study needs to be clearly stated. The authors did not add motivation to their study.
· Using an OLS test to examine the relationship between the SETA program and misuse behaviour is valid. However, it would be useful to mention the sample size and characteristics of the university population surveyed better to understand the research scope and generalizability of the results.
· The authors highlight that based on the results; the SETA program does not directly impact misuse behaviour. This is an important finding, but it would be helpful to discuss the potential implications and reasons behind this outcome briefly. For example, were any limitations or external factors that might have influenced the results?
· The authors briefly mention the importance of perceived cost and information security awareness as psychocognitive variables. Expanding on how these variables were measured and their relevance to understanding students' behaviour would be beneficial. Additionally, briefly discussing the implications of differentiating these variables and treating them as intermediate factors would enhance clarity.
· Advice to Universities Applications is plain. An insightful and critical discussion is expected.
· The abstract needs to be revised, some of the sentences are lengthy. Such as, “In this paper, from the perspective of deterrence theory, we obtain a quantitative study to model the mechanism of influence of SETA program on information system misuse behavior with perceived cost and information security awareness as intermediate variables, using the student population as the research target.” The authors need to be clear, here the of and of make the sentence confusing.
· Besides, the conclusion needs to be rewritten; the words like this paper, this paper need to be avoided.
· several long sentences are hard to follow. The past and present tenses are mixed, which should be unified under a common specification. There are also many typos. The authors should carefully examine the manuscript.
Author Response
Thanks for the reviewer’s comments. We have addressed the problems as follow:
The paper organization (road map) should be clarified for better structure and flow. Currently, the paper organization is missing.
In the first version, we mention and introduce the theories twice and these may make reviewers confused. In this version, we make two parts integrated into part 1 and this will make whole structure more simply and clear.
Our study is based on statistic aspect and we use statistical wordings and structure.
The motivation behind the research study needs to be clearly stated. The authors did not add motivation to their study.
Our original manuscript addressed the issue of school information security pressure in the general environment, which is also one of our motivations. The motivation of the study comes from the thoughts about COVID-19 and we stress the motivation in 1.2 Research Purpose & Significance.
Using an OLS test to examine the relationship between the SETA program and misuse behaviour is valid. However, it would be useful to mention the sample size and characteristics of the university population surveyed better to understand the research scope and generalizability of the results.
Like mentioned before, our questionnaire is publicly available on the web, it is not clear that the specific information about how many people access the questionnaire. The collected 150 sample data can prove that the number of people who see the questionnaire and fill it out. However, since our questionnaire is publicly released on the wjx platform and is not specific to a particular school or region, it cannot give a specific description about the demographic characteristics of the university. At the same time, we believe that a wider sample would make the results more representative.
The authors highlight that based on the results; the SETA program does not directly impact misuse behaviour. This is an important finding, but it would be helpful to discuss the potential implications and reasons behind this outcome briefly. For example, were any limitations or external factors that might have influenced the results?
ISA may be affected by a variety of factors. These factors have little influence alone, but they may have an unpredictable mechanism of influence on ISA when combined. Therefore, this will be a weakness of ours. Our research can only focus on "visible and influential" factors, but cannot reflect various microscopic and fixed factors.
The authors briefly mention the importance of perceived cost and information security awareness as psychocognitive variables. Expanding on how these variables were measured and their relevance to understanding students' behaviour would be beneficial. Additionally, briefly discussing the implications of differentiating these variables and treating them as intermediate factors would enhance clarity.
We have introduced and defined the meaning of perceived cost and information security awareness in part 2 Literature Review and part 3.2 Variable Definition, which provide a clear perception and distinction. Like introducing our questionnaire, we test perceived cost and ISA through several questions related to these two intermediate factors. And we do the reliability test and validity test of the questionnaire to show the rationality.
What’s more, the figure 2 Path Analysis Result demonstrates the mechanism of these two intermediate factors and we amend some wordings on it.
Advice to Universities Applications is plain. An insightful and critical discussion is expected.
Our primary research focuses on SETA, but SETA takes many types. Some organizations use traditional SETA programs, such as organizing training and involving members in training sessions. But some organizations will use new SETA programs, such as incorporating cartoons to control the risk of leaking information. We searched for relevant literature and provide suggestions based on this literature. The literature has shown that the SETA format is important for information security. We have organized and provided our recommendations based on the existing literature.
The abstract needs to be revised, some of the sentences are lengthy. Such as, “In this paper, from the perspective of deterrence theory, we obtain a quantitative study to model the mechanism of influence of SETA program on information system misuse behavior with perceived cost and information security awareness as intermediate variables, using the student population as the research target.” The authors need to be clear, here the of and of make the sentence confusing.
We retained the original content of the abstract, restructured its sentences and expressions to make it more fluent, avoided some inappropriate writing habits, optimized long sentences, and eliminated the problem of confusing sentences.
The conclusion needs to be rewritten; the words like this paper, this paper need to be avoided. several long sentences are hard to follow. The past and present tenses are mixed, which should be unified under a common specification. There are also many typos. The authors should carefully examine the manuscript.
We rewrote the conclusion and revised individual words, double-checking the tense and format of the manuscript.
The references need to be re-arranged in the proper sequence
Done. We have modified the referencing and now it complies with the journal’s guidelines.
Reviewer 3 Report
The paper provides an analysis of the impact of a SETA (Security Education, Training, and Awareness) program on university students' misuse of information systems. It employs a well-structured research methodology, including descriptive statistical analysis, reliability analysis, validity analysis, hypothesis testing using linear regression and path analysis, and provides meaningful findings and insightful discussions. Here is an evaluation and review of the paper:
The paper addresses an important issue concerning the misuse of information systems among university students and investigates the effectiveness of a SETA program in mitigating this behavior. The research focus is well-defined and relevant.
The paper employs a rigorous methodology, including data collection through a questionnaire, statistical analyses (descriptive, reliability, and validity analyses), and hypothesis testing using linear regression and path analysis. These analytical techniques provide a solid foundation for the research findings.
The paper thoroughly examines the impact of the SETA program on perceived cost, information security awareness (ISA), and the reduction of information system misuse. It effectively supports its hypotheses through statistical analyses and provides clear and consistent findings.
The paper recognizes the importance of perceived cost and ISA as psychological cognitive variables and explores their mediating effects between the SETA program and misuse behavior. This enhances the understanding of the underlying mechanisms and strengthens the research model.
The paper provides valuable advice for universities on expanding the SETA program and assessing its effectiveness. The suggestions to enhance program awareness and evaluate its impact contribute to practical application and future improvement.
Although the paper demonstrates quality research, there are several areas where improvements could be made:
- The paper mentions the use of simple random sampling, but additional details on the selection process and representativeness of the sample would strengthen the research design.
- The paper mentions using a questionnaire, it lacks information about the questionnaire design. Including these details would enhance the transparency and reproducibility of the study.
- The paper mentions that ISA levels are influenced by various factors, but a more extensive discussion of potential confounding variables and limitations of the study would provide a more balanced perspective.
- at line 441 („5.2. Descriptive Statistical Analysis”) at the end of the page has only a title, it should be repositioned. The same goes for line 508 („6.1.2. The Impact of SETA Program on Perceived Costs”)
Author Response
Thanks for the reviewer’s comments. We have addressed the problems as follow:
The paper mentions the use of simple random sampling, but additional details on the selection process and representativeness of the sample would strengthen the research design.
We include a more detailed description of the simple random sampling process and sample representativeness in Subsection 5.2. Since our questionnaire is publicly available on the web, it is not clear that the specific information about how many people access and browse the questionnaire. The collected 150 sample data can prove that the number of people who see the questionnaire and fill it out. Besides, we also provide a more detailed description of the group of people who received the questionnaire, which covers not only all educational stages but also schools in different regions. This minimizes the error and further supports the fact that we collected a representative sample.
The paper mentions using a questionnaire, it lacks information about the questionnaire design. Including these details would enhance the transparency and reproducibility of the study.
We have included a description of the questionnaire design in subsection 5.1, specifically including how the questions in each section are conceptualized and what literature is referenced to demonstrate the validity of the questions.
The paper mentions that ISA levels are influenced by various factors, but a more extensive discussion of potential confounding variables and limitations of the study would provide a more balanced perspective.
Information security awareness encompasses many dimensions, like an individual's level of knowledge, time spent in contact with information systems, and even the attitudes of those around the person can affect information security awareness. This paper focuses on the SETA program, and an extensive discussion of information security awareness would lead to a detachment from the main idea of the study. Therefore, we chose to summarize several theories we covered briefly and to focus on empirical research.
At line 441 („5.2. Descriptive Statistical Analysis”) at the end of the page has only a title, it should be repositioned. The same goes for line 508 („6.1.2. The Impact of SETA Program on Perceived Costs”)
We have reviewed whole wordings and amended accordingly.
Round 2
Reviewer 2 Report
The authors have addressed my comments. I don't have further comments.